# Development and Application of IoT Monitoring Systems for Typical Large Amusement Facilities

**DOI:** 10.3390/s24144433

**Published:** 2024-07-09

**Authors:** Zhao Zhao, Weike Song, Huajie Wang, Yifeng Sun, Haifeng Luo

**Affiliations:** 1Key Laboratory of Special Equipment Safety and Energy-Saving for State Market Regulation, China Special Equipment Inspection and Research Institute, Beijing 100029, China; ouczhaozhao@163.com (Z.Z.);; 2China Special Equipment Inspection and Research Institute, Beijing 100029, China; 3School of Technology, Beijing Forestry University, Beijing 100083, China

**Keywords:** internet of things, sensors, large amusement facilities, equipment monitoring

## Abstract

The advent of internet of things (IoT) technology has ushered in a new dawn for the digital realm, offering innovative avenues for real-time surveillance and assessment of the operational conditions of intricate mechanical systems. Nowadays, mechanical system monitoring technologies are extensively utilized in various sectors, such as rotating and reciprocating machinery, expansive bridges, and intricate aircraft. Nevertheless, in comparison to standard mechanical frameworks, large amusement facilities, which constitute the primary manned electromechanical installations in amusement parks and scenic locales, showcase a myriad of structural designs and multiple failure patterns. The predominant method for fault diagnosis still relies on offline manual evaluations and intermittent testing of vital elements. This practice heavily depends on the inspectors’ expertise and proficiency for effective detection. Moreover, periodic inspections cannot provide immediate feedback on the safety status of crucial components, they lack preemptive warnings for potential malfunctions, and fail to elevate safety measures during equipment operation. Hence, developing an equipment monitoring system grounded in IoT technology and sensor networks is paramount, especially considering the structural nuances and risk profiles of large amusement facilities. This study aims to develop customized operational status monitoring sensors and an IoT platform for large roller coasters, encompassing the design and fabrication of sensors and IoT platforms and data acquisition and processing. The ultimate objective is to enable timely warnings when monitoring signals deviate from normal ranges or violate relevant standards, thereby facilitating the prompt identification of potential safety hazards and equipment faults.

## 1. Introduction

Entertainment attractions, including theme parks, water parks, and carnivals, constitute an essential component of the tourism industry worldwide. The safety of large amusement facilities, as the cornerstone of these attractions, is paramount to public safety. Globally, ride malfunctions are not rare; in fact, 182 incidents, including 51 fatal ones, were reported in 38 countries within a single year [1]. There is an urgent need for proactive identification of safety hazards to prevent future accidents. Currently, safety checks for large amusement facilities rely on periodic inspections conducted either by manual or automated methods [2], lacking a smart sensor network and IoT platform for real-time monitoring of equipment signals.

Advancements in wireless communication and microelectronics, coupled with cost reductions, have facilitated the widespread application of sensor networks in numerous innovative fields [3]. These networks are utilized in smart homes, military surveillance, traffic control, environmental monitoring, and other applications. Sensor integration within IoT platforms has revolutionized environmental interaction and management by enabling efficient data collection, remote monitoring, and automated control [4,5,6,7]. For instance, the rapid progress of sensors and the IoT has presented new opportunities for agriculture, allowing timely access to production information, and transforming agricultural processes to ensure high yields and environmentally friendly practices [8,9,10]. Its primary goal is to merge past analytical data with current monitoring data to offer more accurate models and best solutions, thereby striving for sustainable smart agriculture in the future. Additionally, in the automotive and transportation domain, the IoT has been tested for various in-vehicle functions [11], monitoring parameters such as air quality, traffic flow, weather conditions, and signal strength. Simplified algorithms and intuitive data visualizations enable informed decision-making by drivers, authorities, and urban planners [12,13,14,15]. Meanwhile, intelligent healthcare emerges as a promising IoT application, remotely delivering medical services, reducing costs, and enabling independent living for patients. These systems also aid in early disease detection, significantly enhancing patients’ quality of life [16,17]. Future healthcare architectures will harness VR, deep learning, big data, and high-performance computing for reliable cloud/edge/fog networking solutions [18,19,20,21,22]. The advancement of the IoT and modern sensors has transformed environmental monitoring into a truly intelligent system [23]. For instance, water quality is monitored through digital image analysis and machine learning [24], while air quality assessments are conducted via fixed and mobile sensor nodes [25]. The data captured by these intelligent sensor nodes is then processed and analyzed with the aid of machine learning techniques. For example, Singh [26] has introduced an intelligent water quality monitoring framework that incorporates AI sensors to predict indicators without the need for electronic sensors; specifically, employing machine learning algorithms to forecast the concentration of E. coli in water. In addition, a novel machine learning-based IoT system has been developed to track an individual’s stress level and movement velocity during physical activities by examining factors such as body temperature and sweat [27]. Capitalizing on wireless sensor networks, machine learning techniques, and IoT methodologies, the detection and classification of pollutants in aquatic environments have been effectively executed [28].

Currently, due to the unique complex structure and diverse operational characteristics of amusement facilities, a dedicated IoT monitoring system for them has not yet been developed. Herein, inspired by the smart monitoring systems enabled by the IoT in various industries, this paper presents a customized sensor-driven IoT condition monitoring system specifically designed for large amusement facilities. Focusing primarily on roller coasters, we conducted an analysis of past accidents to pinpoint prevalent faults and their causes. Based on this analysis, we selected seven distinct sensor types: audio–video surveillance, photoelectric sensors, acceleration sensors, speed sensors, vibration sensors, electrical signal sensors, and stress sensors. These sensors provide real-time monitoring of crucial parameters, encompassing structural deformations, damage, unauthorized operations, ride smoothness, power system jamming and wear, control system anomalies, and structural overloads. Additionally, an integrated data processing and transmission module ensures real-time information relay to a designated IoT platform. Due to the complex and rapid changes in roller coaster movements, all sensor connections to the IoT platform in this study are wireless to ensure passenger and equipment safety. By analyzing the collected physical signals, we facilitate intelligent safety management and early fault warnings specifically tailored for roller coasters. This approach surpasses the limitations inherent in traditional periodic manual inspections, which often suffer from delays in issue identification due to limited resources and expertise. Therefore, our proposed sensor network and IoT platform facilitate the early detection of abnormal signals in roller coaster operations that deviate from normal ranges or violate national or regional standards. This enables the prompt implementation of maintenance and repair measures to ensure the safety of roller coaster operations. The innovations of this paper are as follows:Identification of the required signal types for monitoring the operational status of roller coasters, based on a comprehensive analysis of numerous past failure cases.Development of a specialized monitoring sensor system tailored to the unique structural and operational characteristics of roller coasters.Design of an informative IoT platform for large-scale amusement facilities, featuring the visualization of monitoring data to enhance operational safety and efficiency.

## 2. Analysis of Health Monitoring Indicators

The large roller coaster serves as a prototypical large amusement facility, lifted by a dynamic system to the highest point of the track. Once there, gravitational potential energy propels it through a series of rolls and spiral motions along the rigid track. Its thrill-inducing experience has earned it immense popularity among amusement seekers. However, any malfunction in its components could lead to severe failures or accidents. Therefore, it is imperative to devise a monitoring system specifically tailored to the unique operational characteristics of roller coasters.

The development of a condition monitoring system for large roller coasters necessitates balancing monitoring effectiveness and cost, making it essential to conduct failure analysis to identify potential risk areas. This analysis lays the foundation for sensor selection, data processing terminal design, and data analysis. Figure 1a presents a systematic analysis of roller coaster failure cases over the years. In terms of failure locations, the primary failures observed relate to car bodies (cracks in the frame and wheel mounts), safety-lever locking devices (failure of locking cylinders or structural damage or deformation), tracks (weld cracks, missing or broken connecting bolts), control systems (failure of safety functions and illegal operation), axles (cracks, corrosion, abnormal wear), and failure of anti-reversal lifting devices. Figure 1b demonstrates that large roller coaster failures primarily concentrate on three main load-bearing components: the car body, track, and axles, accounting for 29% of the total failure cases. The primary failure mode observed in these components is cracking, partially due to the high speed and impact forces experienced by roller coasters. Under alternating loads and impacts, these load-bearing components are susceptible to fatigue cracking. Secondly, damage and deformation in the passenger carrying system, including the cabin, safety bars, and seatbelts, accounts for 17% of failure cases. Additionally, illegal operation and transmission system malfunction, encompassing anti-reversal devices, braking systems, and collision prevention mechanisms, constitute 15% and 14% of failure cases, respectively. Furthermore, electrical control systems significantly contribute to the failure cases, representing 12% of the total. Failures in the electrical control system encompass system circuit breaks, sensor malfunction resulting in no signal, and the failure of interlocking protection mechanisms. Based on the failed parts and causes depicted in Figure 1, sensors can be accurately positioned at crucial failure points of essential components for real-time monitoring. Thereafter, the diverse monitoring data can be processed and uploaded to the IoT platform, facilitating a comprehensive diagnosis of the roller coaster’s operational health status. 

## 3. Architecture of IoT Monitoring System

Leveraging the extensive coverage of communication networks and the widespread adoption of sensing technology, the IoT enables real-time, comprehensive monitoring of roller coaster operational status by collecting enormous amounts of data. Typically, the IoT architecture comprises three layers: the perception layer, network layer, and application layer. The perception layer encompasses diverse sensors that gather a wide range of physical information during the roller coaster’s operation. The network layer is tasked with transmitting the sensed data and facilitating connectivity with the application layer servers. The application layer employs cloud-based services that engage users, facilitating high-quality intelligent services encompassing data integration, analysis, and computation. The IoT system architecture designed for monitoring and assessing the health of roller coasters is schematically represented in Figure 2. Wi-Fi-enabled wireless sensors communicate locally via TCP/IP, forming a unified LAN with nearby access points. The sensor signals within the LAN are collected through Data Acquisition and then uniformly sent to the cloud server on the internet via the Processing Terminal. The 4G-capable sensors communicate directly with the cloud.

Based on the analysis of failure modes depicted in Figure 1, the requirements for roller coaster monitoring indicators have been summarized. Seven primary monitoring indicators have been identified, specifically including audio and video, operating cycles and speed, acceleration, vibration, electrical signals, and stress. Depending on the specific design of the roller coaster, hardware systems tailored to different monitoring indicators have been selected. These systems encompass sensors, data acquisition, processing, and transmission terminals. The sensors employed for the corresponding monitoring indicators include surveillance cameras, photoelectric sensors, acceleration sensors, vibration sensors, intelligent circuit breakers, and stress sensors.

### 3.1. Audio and Video Surveillance

Audiovisual monitoring of roller coasters plays a pivotal role in ensuring smooth operation and rider safety. This technology combines video surveillance with audio recording, providing an overview of the coaster’s status and behavior, as illustrated in Figure 3. The mechanism of audiovisual monitoring involves the deployment of high-resolution cameras and audio sensors (see Table 1) at strategic points along the roller coaster track. These cameras capture real-time footage of the coaster’s movement, enabling maintenance staff to visually assess its running smoothness and detect any malfunctions. Audio sensors, meanwhile, record sounds emitted by the coaster and its components, providing early warning information through the detection of abnormal sounds during operation. Cameras and audio sensors are installed at critical locations, such as turns, drops, and intersections, to ensure maximum coverage. These devices are then connected to a centralized monitoring station, where operators can access live feeds and recorded data.

The advantages of audiovisual monitoring are numerous. Firstly, it allows operators to visually assess the coaster’s running smoothness, identifying any unexpected behaviors that may indicate a problem. Secondly, this system provides information about the coaster’s surroundings, enabling operators to promptly detect any obstacles or unauthorized individuals in the designated operational area. This is crucial for preventing collisions and ensuring the safety of riders and personnel. Furthermore, audiovisual monitoring can record interactions between operators and roller coasters, ensuring that operators adhere to safety protocols and maintain proper operational procedures. Lastly, in the event of a fault or accident, audiovisual monitoring provides invaluable evidence for investigation. Recordings of incidents can be analyzed to determine their causes and identify any contributing factors. This information is crucial for preventing future occurrences and improving the safety of roller coasters.

### 3.2. Photoelectric Sensor

In the realm of roller coaster operation frequency and speed monitoring, the utilization of photoelectric reflection sensors (Table 2) has emerged as a viable and effective solution. These sensors are designed to emit a signal whenever the roller coaster obstructs their path, enabling precise tracking of the coaster’s movements.

One of the primary monitoring metrics is the number of operational cycles, which are typically counted from the coaster’s exit from the station to its return. To accurately measure this, sensors are positioned at both the exit and entry points, allowing for a seamless count of each cycle. The significance of this metric lies in its ability to assess fatigue-related damage, which often manifests as cracks caused by prolonged exposure to alternating loads. Such damage can be effectively mitigated through regular monitoring and timely maintenance.

Another crucial monitoring metric is the coaster’s speed, particularly at its peak, which usually occurs at the lowest point of descent as potential energy converts fully into kinetic energy, as shown in Figure 4. To measure this, two closely positioned photoelectric sensors record the time it takes for the roller coaster to pass between them. This data then calculates the maximum instantaneous speed, revealing valuable insights into its performance and safety. Any speed deviations or irregularities during operation can potentially indicate underlying issues, such as abnormal friction resulting from wheel frame and track deformation or insufficient lubrication. Recognizing these problems promptly aids management in taking swift corrective actions.

### 3.3. Acceleration Monitoring System

The acceleration of a roller coaster is a crucial parameter for analyzing its operational status. The incorrect location of a roller coaster can be determined based on atypical data timing. For instance, roller coaster manufacturers can identify defects in specific sections of the track by examining abnormal values in on-board acceleration and wheel frame stress, thereby optimizing the track design. Additionally, roller coaster tracks typically consist of various elements, such as loops, spirals, and straight sections. High acceleration can increase the risk of passenger injury. To ensure passenger safety, acceleration must be kept within a certain range. Therefore, continuous acceleration monitoring during roller coaster operation is essential.

The reference point for measuring acceleration is generally located 600 mm above the seat, which approximates the position of the human heart. Figure 5a illustrates the installation of the acceleration system on the roller coaster. The direction towards the passenger’s back is defined as the positive *x*-axis, the direction towards the passenger’s left hand is defined as the positive *y*-axis, and the seat’s downward direction is defined as the positive *z*-axis. In Figure 5b, this paper presents the acceleration monitoring results from a roller coaster operating continuously for three cycles. Determining whether acceleration values fall within the allowable range can be challenging, as they depend on duration. Therefore, referencing the Chinese standard GB-8408 (2018) [29], this study has selected three representative durations (Δ*t* = 0.05 s, 0.1 s, and 0.2 s) to depict the acceptable regions for combinations of lateral acceleration (*a*_y_) and vertical acceleration (*a*_z_) that significantly impact human health, as shown in Figure 5c. The representative durations of the allowable combined acceleration values are consistent with EN13814-2019 [30]. The results reveal that, with a 0.01 s sampling interval, there are zero, six, and eight consecutive data points surpassing the thresholds of 0.05 s, 0.1 s, and 0.2 s, respectively. From Figure 5c, the criteria for combined acceleration are as follows:

If (*n*_1_ + 1) Δ*t* > 0.2, the acceleration does not meet the standard requirements.

If (*n*_2_ + 1) Δ*t* > 0.1, the acceleration does not meet the standard requirements.

If (*n*_3_ + 1) Δ*t* > 0.05, the acceleration does not meet the standard requirements.

Otherwise, the acceleration meets the standard requirements. 

**Figure 5 sensors-24-04433-f005:**
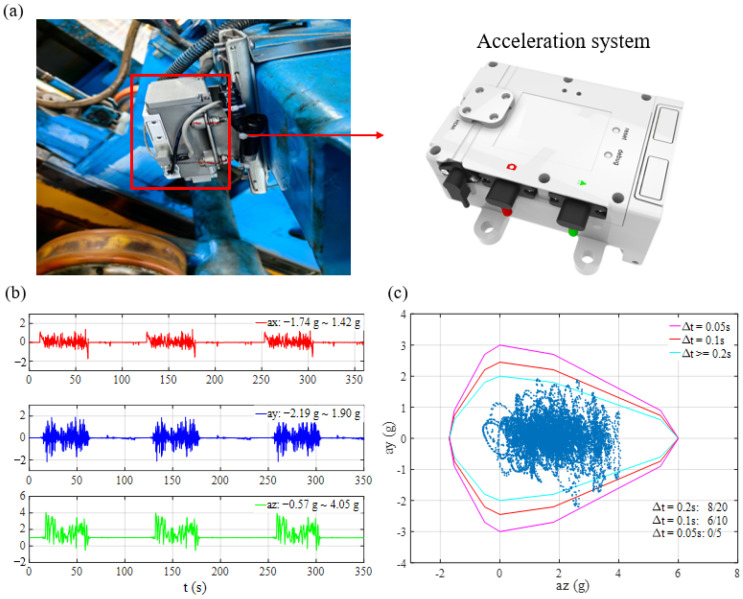
(**a**) Schematic illustrating the installation of an acceleration monitoring system on a roller coaster; (**b**) representation of anisotropic acceleration; and (**c**) combined acceleration profiles over three operational cycles of the roller coaster.

Therefore, when the sampling interval Δ*t* is less than 0.01 s, more points exceeding the limits are allowed if above criteria met. The computed durations (Δ*t*) remain under these thresholds, thereby complying with the standard criteria.

The wireless acceleration monitoring system comprises wireless acceleration sensor nodes, a wireless gateway, and a computer equipped with dedicated data acquisition and processing software (Matlab 2016). The system parameters are listed in Table 3. Each wireless acceleration sensor node primarily consists of four modules: power supply, acceleration sensor, data acquisition and processing, and wireless communication. Initially, the power supply module provides electricity to the node. Subsequently, the acceleration sensor transforms force signals into electrical signals. These signals are then amplified and filtered by the data acquisition and processing module, which sends them to an analog-to-digital converter for digitization. The digitized signals are then forwarded to the main processor for further digital signal processing, enabling the calculation of the sensor’s effective value, among other computations. The wireless communication module employs the IEEE 802.15.4 [31] standard-based wireless protocol for data transmission and its acceptable latency satisfies the requirements for roller coaster acceleration monitoring. This is because the acceleration data from the previous round is typically analyzed after the vehicle arrives at the station to determine whether to continue operation. The wireless gateway receives acceleration data from the sensor nodes and relays it to the computer via a USB interface for storage. The data acquisition software gathers real-time data from multiple sensor nodes, simultaneously displaying, analyzing, and storing this information.

### 3.4. Speed Monitoring System

An essential dynamic performance parameter of roller coasters is operating speed, as this serves as a vital indicator for evaluating the ride’s stability during inspections. Abnormal speeds might indicate problems such as loose wheel frames or unusual deformations in the roller coaster structure. Furthermore, the operating speed determines whether the roller coaster can ascend to its peak without slipping backwards. However, roller coaster designs often lack built-in speed detection devices, making it impossible to directly measure the operating speed. Presently, inspectors rely on handheld radar speed guns for segmented speed tests, but this method faces challenges such as instability, reduced accuracy, and sensitivity to the inspector’s hand position. Additionally, it does not allow for continuous speed monitoring throughout the ride. Given the demanding operating environment of roller coasters and their high speeds, this paper opts for a specially designed millimeter-wave radar as the speed monitoring device. This radar is mounted on the roller coaster track column at the lowest point of the dive (refer to Figure 6a), where the roller coaster attains its maximum speed. To prevent collisions with the station due to excessive speed, an additional speed monitoring device is positioned before the roller coaster returns to the station for braking.

Table 4 presents the detailed parameters of the sensor, which has high accuracy, robust anti-interference capability, extended detection distance, and broad detection range. This sensor can operate seamlessly even under severe weather conditions. The millimeter-wave radar sensor serves as the backbone sensing component in our speed monitoring device, as depicted in Figure 6b,c. For electromagnetic wave signal processing, we utilize multi-level digital frequency modulation (MFSK). This approach combines the advantages of frequency-shift keying (FSK) and linear frequency modulation continuous wave (LFMCW), effectively addressing LFMCW’s speed and distance matching challenges, as well as the low-resolution issues associated with FSK due to its reliance on the Doppler shift and same-speed targets. Consequently, this enhances the product’s stability and reliability. When it comes to intermediate frequency signal processing, we employ an enhanced FFT algorithm, refining its frequency through chirp-z transformation. This boosts the precision of frequency measurement. By integrating millimeter-wave radar sensors, edge computing gateways, and a health management cloud platform specifically designed for amusement parks and scenic spots with manned equipment, we achieve precise monitoring of the roller coaster’s maximum operating speed and return speed. Figure 6d illustrates the speed monitoring values recorded during one complete operating cycle of the roller coaster. In case the speed exceeds the safe threshold range, immediate alarms are triggered, enabling prompt maintenance measures to ensure uninterrupted and safe operation.

### 3.5. Vibration Monitoring System

Gears, shafts, and rolling bearings are essential components of roller coaster transmission systems and are responsible for up to 90% of failures. Furthermore, when these components malfunction, they can mutually influence each other, resulting in compounded failures. Therefore, analyzing the primary failure modes of these critical components is crucial for accurate fault diagnosis. For instance, in a roller coaster’s transmission mechanism, shafts are interconnected by couplings. During operation, issues such as shaft misalignment, imbalance, or bending can subject the rotating shaft to significant radial alternating forces, causing vibration. Additionally, failures in gears and bearings attached to the shaft can contribute to shaft malfunction. Compared to other indicators, vibration characteristics provide a swift, comprehensive, and accurate reflection of equipment status. They also reveal the nature and extent of faults in gears, rolling bearings, and shaft systems. Hence, to prevent potential mechanical failures in the roller coaster transmission system, it is imperative to install vibration sensors and terminals for real-time vibration signal acquisition and storage. This facilitates prompt early warnings in response to abnormal vibration signals.

In this study, MEMS triaxial acceleration sensors ADXL355 serve as the core sensing elements, and their specific parameters are detailed in Table 5. Four sensor probes are strategically placed on the lift chain drive unit of the roller coaster; the assembly of one such probe is depicted in Figure 7a. The captured signals undergo empirical mode decomposition (EMD) followed by Hilbert transform analysis of the primary components. The raw data are first analyzed and processed through an edge computing gateway, subsequently transmitted to a cloud platform via a 4G wireless signal, facilitating additional data processing on the cloud. By harnessing the cloud platform for amusement equipment health management, the data are then presented visually. Through the analysis of the time domain diagrams of the vibration signals (refer to Figure 7b), we achieve precise monitoring of the roller coaster transmission system’s vibration signals, thereby ensuring its smooth and healthy operation.

### 3.6. Intelligent Circuit Breaker

The electrical system of a roller coaster typically comprises a power supply, control system, drive motor, sensors, and various safety devices. These components collaborate to ensure the roller coaster operates smoothly along a predetermined path at a designated speed. However, electrical failures occasionally occur due to the electrical system’s complexity and the influence of external environmental factors. Instability in voltage and current can directly impact the performance of the roller coasters drive motor and control system. Consequently, the motor might fail to deliver adequate power, leading to speed fluctuations during operation. This can result in the roller coaster failing to clear the peak and colliding backwards into the station. Additionally, unstable voltage and current can cause circuit overloads and short circuits, particularly in harsh environments characterized by high temperatures and humidity, potentially sparking electrical fires.

Therefore, monitoring the electrical system signals of roller coasters is significant. Conventional electrical signal monitoring primarily involves using current transformers to monitor motor current signals, which may not fully represent the electrical system’s status. In this study, we utilize a rail-mounted three-phase multifunctional power monitoring terminal (refer to Table 6). With its convenient rail installation method, this terminal can be effortlessly installed alongside air switches, circuit breakers, and contactors (see Figure 8a). It measures 30 electrical parameters, including current, voltage, active power, reactive power, and four-quadrant electrical energy on the three-phase power grid. These parameters are subsequently transmitted in real-time to the system data center via a 4G network. Instability in three-phase voltage and current is a significant contributor to electrical failures in roller coasters. This study presents the monitoring outcomes for three-phase current and voltage signals (illustrated in Figure 8b). Through meticulous data analysis, it becomes feasible to assess whether the three phases are balanced, detect anomalies in a specific phase during operation, and comprehensively evaluate the insulation status of the motor windings. This facilitates early detection and warning of electrical failures in roller coasters.

### 3.7. Stress Monitoring System

During the high-speed dive and roll of the roller coaster, material strength failure may occur in locations where the impact between the car body and the track is significant, due to local stress concentration. Additionally, long-term alternating loads can also cause structural fatigue damage. Therefore, it is imperative to monitor the stress at critical structures and specific locations along the roller coaster track. Traditional stress–strain testing techniques cannot accommodate the complex operating conditions of roller coasters, which are characterized by variable structural loads, long transmission distances, and limited installation spaces. Consequently, this study utilizes a wireless stress–strain testing system to monitor the stress in critical structures of the roller coaster, with detailed parameters outlined in Table 7. This advanced system is a data acquisition setup based on wireless strain sensor nodes, comprising primarily of data acquisition software, wireless strain nodes, strain gauges, and a wireless gateway. The wireless stress–strain testing system eliminates the need for cumbersome wiring and utilizes wireless digital signal transmission to reduce the noise interference often associated with long cable runs. As a result, the entire measurement system demonstrates high measurement accuracy and anti-interference abilities. The core principle of this system lies in converting resistance changes in the strain gauges into voltage signals via wireless strain sensor nodes. These signals are then seamlessly transmitted to a computer, where they undergo processing and analysis using specialized software.

In this study, stress monitoring was carried out at two specific test points: the junction of the roller coaster track (refer to Figure 9a) and a crucial wheel-frame structure (see Figure 9b). We present the stress monitoring outcomes from three successive operational cycles of the roller coaster operating under full load conditions. In terms of stress monitoring, the main load-bearing components of the car body, track, and column structure are typically the focus of attention. These structures are generally made of carbon steel, such as Q355 steel and 20# steel. Since there is little difference in mechanical properties between these steel grades, the same safety factor of 3.5 is selected according to the requirements of the mandatory Chinese national safety standard GB8408–2018 [29]. Hence, if the monitored stress levels surpass the prescribed design safety factor in the stress monitoring data pertaining to the roller coaster’s structure and track, the IoT platform will instantly receive alerts regarding the excessive stress values and their precise locations. In summary, wireless stress monitoring technology has established itself as a reliable testing method that facilitates wireless signal transmission and overcomes the limitations inherent in traditional stress–strain testing approaches, especially in the context of roller coaster evaluations.

### 3.8. IoT Platform

The IoT platform primarily focuses on facilitating user interaction and data presentation and consists mainly of two key interfaces: the system homepage (refer to Figure 10a) and the equipment monitoring interface (see Figure 10b). The system homepage encompasses three main sections. Firstly, the data statistics section located at the top presents details such as the number of user units, equipment quantity, and manufacturers. Secondly, the central portion of the interface displays a map depicting the distribution of equipment installed throughout China. Lastly, the real-time monitoring data statistics, positioned on both sides of the interface, encompassing the number of IoT-connected devices, total equipment operating hours, health status data, fault warning information, and message notifications. By utilizing this homepage, users can gain a comprehensive understanding of the distribution of large amusement facilities in China, the layout of the internet of things devices, and the safety status of associated equipment.

The equipment monitoring interface, serving as the cornerstone of the monitoring system and the primary medium for displaying on-site sensor data, comprises seven distinct modules. These modules encompass a wide range of functions, from presenting fundamental equipment details such as device information, technical parameters, and unit specifics, to showcasing real-time surveillance footage and audio. Additionally, they display core equipment status, including IoT connectivity, environmental conditions, and GPS coordinates, while also providing insights into daily operational durations and usage patterns. Furthermore, the interface delivers real-time equipment health assessments based on diverse sensor data, alerts users to any status changes, and exhibits up-to-date sensor data through real-time monitoring indicators. This study selected Happy Valley in Chengdu, China, as a pilot site and conducted online monitoring applications on four large roller coasters in the park: the Dive coaster (Bolliger & Mabillard Consulting Engineers Inc., Burgdorf, Switzerland), the Wooden coaster (Great Coasters International, Inc., Kentucky, IL, USA), the MEGA coaster (Intamin Transportation Ltd., Vaduz, Liechtenstein), and the Mine Train (Vekoma Rides Manufacturing Bv, Vlodrop, the Netherlands). When the monitoring signals exceed the normal fluctuation range or fail to meet the standard requirements, the IoT platform will push information and alerts to users based on the time and location of the abnormal data. For instance, when the stress and acceleration do not comply with the regulations specified in GB8408 [29], or there are abnormal increases and decreases in vibration signals and electrical signals, the IoT platform will issue reminders for abnormal data, thereby assisting personnel in inspecting the equipment. 

Utilizing the 5 GHz Wi-Fi band, this study enhances network capacity during peak hours through its increased channels and bandwidth. The monitoring data are presented based on the operational rounds of the roller coaster, ensuring data display for each round only after completion. This approach grants the system a relatively high tolerance for latency, enabling data to be showcased on terminals post round completion despite potential delays during peak periods. Based on the various test data, the sensors have varying upload frequencies. During the operation of the equipment, sensors such as electrical signals, vibration, and acceleration upload data once per second. For instance, sensors monitoring operating speed will upload data when a detection event occurs. The system employs IoT communication technology based on the MQTT protocol, ensuring data upload under weak network conditions. Currently, the WiFi and 4G network communication used can meet the requirements for sensor data transmission in roller coaster operation monitoring. The availability of these seven monitoring modules in real-time, coupled with the option to download them for deeper data analysis, significantly elevates the efficacy of equipment safety measures, enables prompt fault warnings, and ultimately enhances the operational safety of the equipment.

## 4. Conclusions

In summary, we have developed an online monitoring system for large amusement rides, leveraging sensor networks and the IoT framework. Taking the roller coaster as a prime example of such rides, we have implemented online tracking of vital physical parameters critical to operational safety, drawing insights from past accident case studies. Our approach incorporates a camera-based monitoring system for audio and video capture, thereby assisting managers in obtaining a real-time understanding and documentation of the roller coaster’s operational status. Photoelectric sensors keep track of the operation count and peak speeds, while acceleration and speed sensors monitor the roller coaster’s stability on the track. Vibration sensors detect common mechanical failures in the drive system, intelligent circuit breakers identify abnormal signals within the equipment control system, and stress sensors monitor stress levels in high-impact zones of the roller coaster and track, preventing material failures due to excessive force. All the collected physical data converges on our custom-developed IoT online platform, enabling staff to view the real-time operational status of the amusement rides via computers and mobile devices. In the forthcoming work, the assessment of data through thresholds and applicable standards will be complemented by the employment of artificial intelligence algorithms as a pivotal technique for identifying potential anomalies in fluctuating data, thereby enhancing the fault diagnosis process. This system has the potential to replace conventional manual periodic inspections, effectively tackling the challenge of limited human experience and energy in timely detecting equipment hazards. This innovation offers an intelligent real-time monitoring IoT platform, bolstering the safety of large amusement rides and safeguarding both the equipment and its passengers. Moreover, the platform amasses a wealth of equipment monitoring data, and this valuable big data can benefit other segments of the amusement industry, such as sales, research, and maintenance.

## Figures and Tables

**Figure 1 sensors-24-04433-f001:**
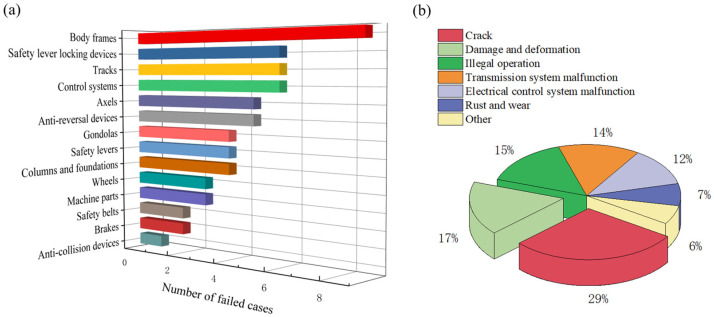
(**a**) Statistics regarding failed parts and (**b**) various modes of large roller coasters.

**Figure 2 sensors-24-04433-f002:**
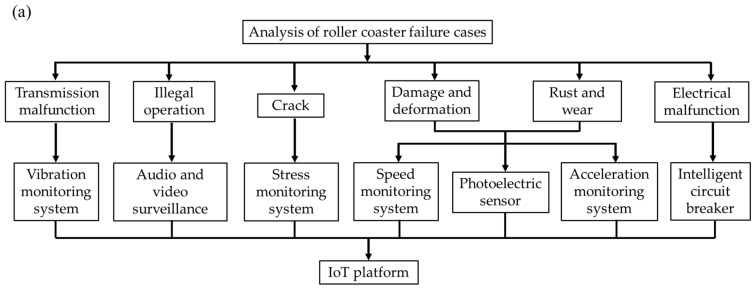
(**a**) The flow chart and (**b**) the architecture of the IoT system designed for monitoring roller coasters.

**Figure 3 sensors-24-04433-f003:**
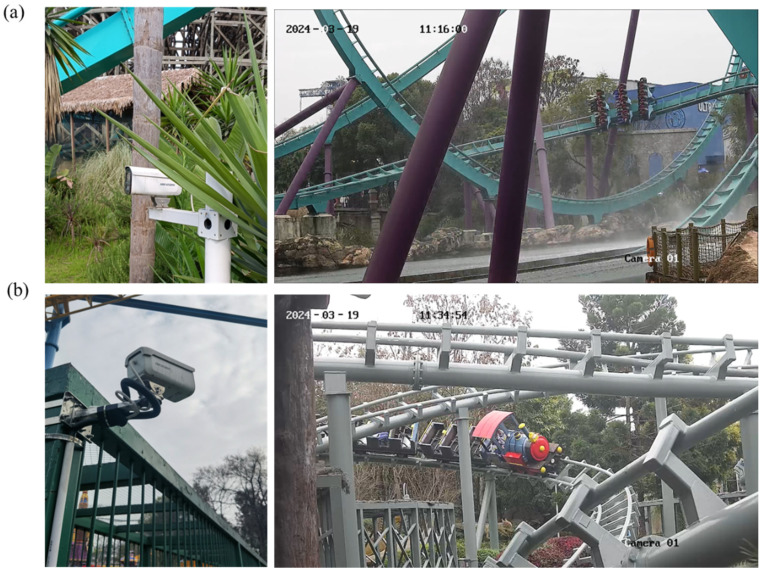
Monitoring interfaces displaying real-time screens of (**a**) the “Dive Coaster” and (**b**) the “Mine Train”.

**Figure 4 sensors-24-04433-f004:**
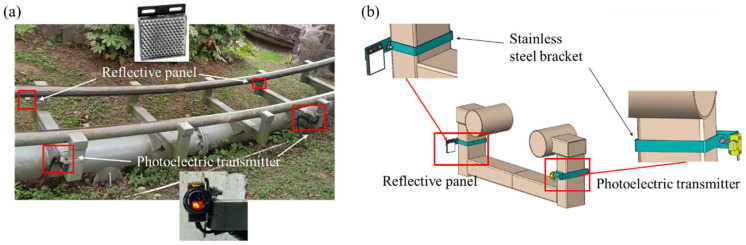
(**a**) Installation diagram and (**b**) schematic of photoelectric sensors utilized for on-site measurement of run counts and speed.

**Figure 6 sensors-24-04433-f006:**
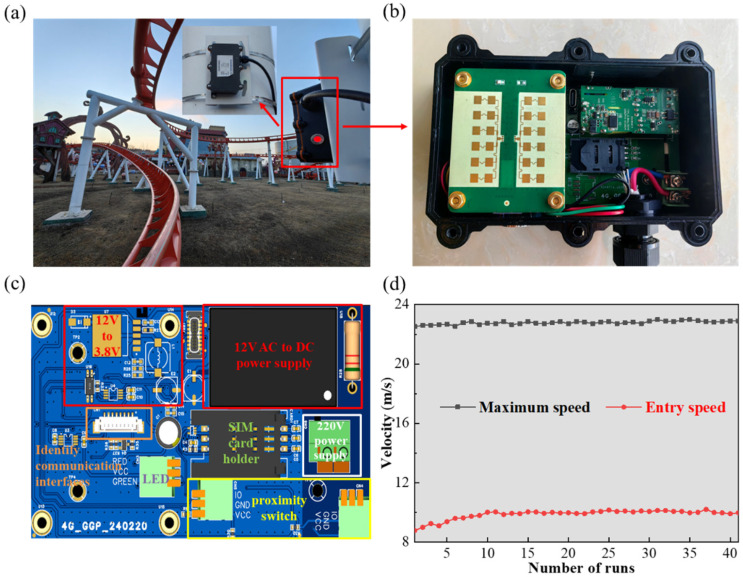
(**a**) Installation of the speed monitoring system on a roller coaster track column. (**b**) Schematic diagram, (**c**) integrated electrical components of the speed monitoring system, and (**d**) results exhibiting maximum speed and speed upon entering the station.

**Figure 7 sensors-24-04433-f007:**
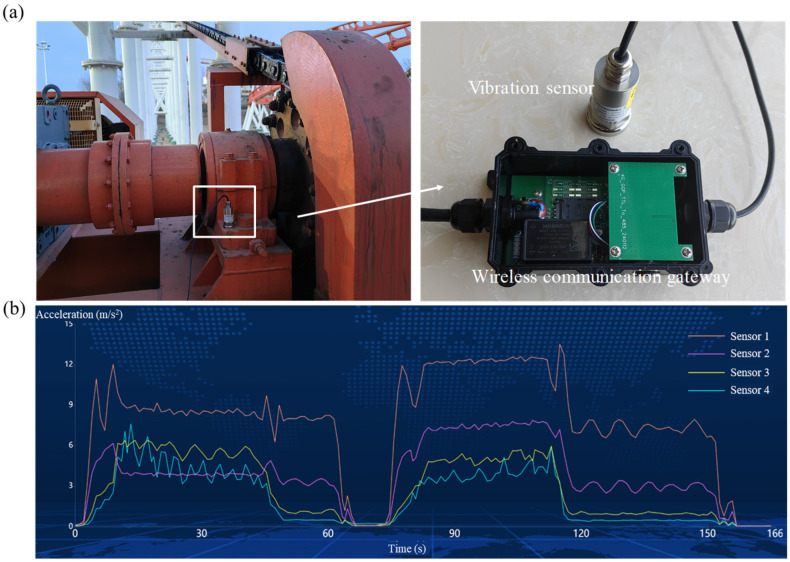
(**a**) Installation of the vibration monitoring system on the lift chain drive. (**b**) Acceleration data captured by four vibration sensors during a single roller coaster operation cycle.

**Figure 8 sensors-24-04433-f008:**
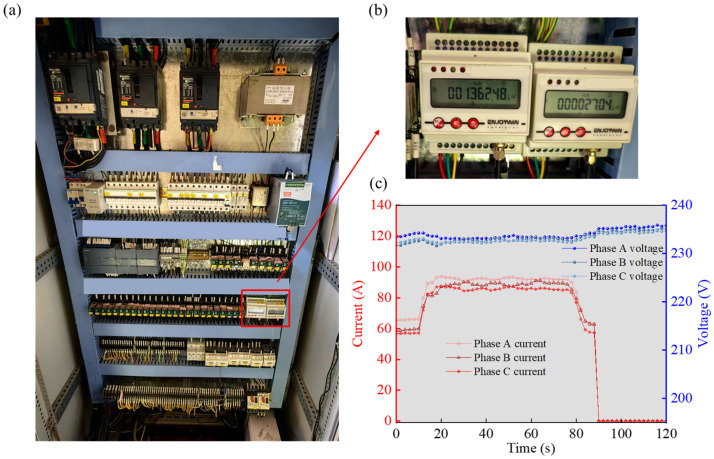
Diagrams of (**a**) the power control cabinet and (**b**) the intelligent circuit breaker. (**c**) Recordings of three-phase voltage and three-phase current during a single operation cycle.

**Figure 9 sensors-24-04433-f009:**
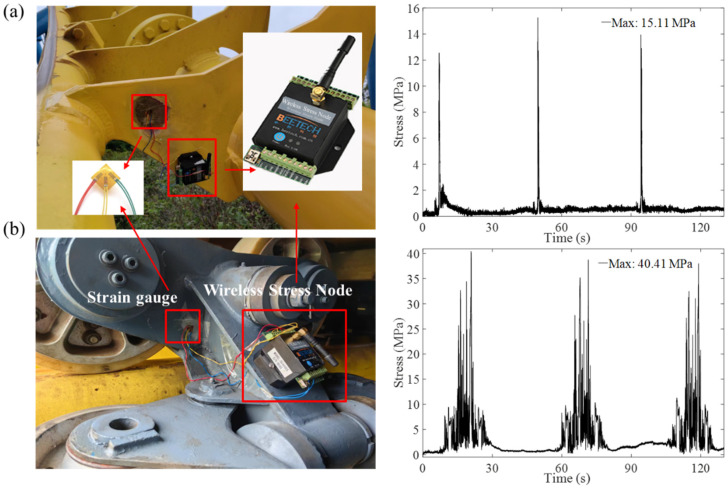
Wireless acceleration monitoring system installed on (**a**) the track and (**b**) the wheel frame of a roller coaster, along with acceleration data recorded over three operational cycles.

**Figure 10 sensors-24-04433-f010:**
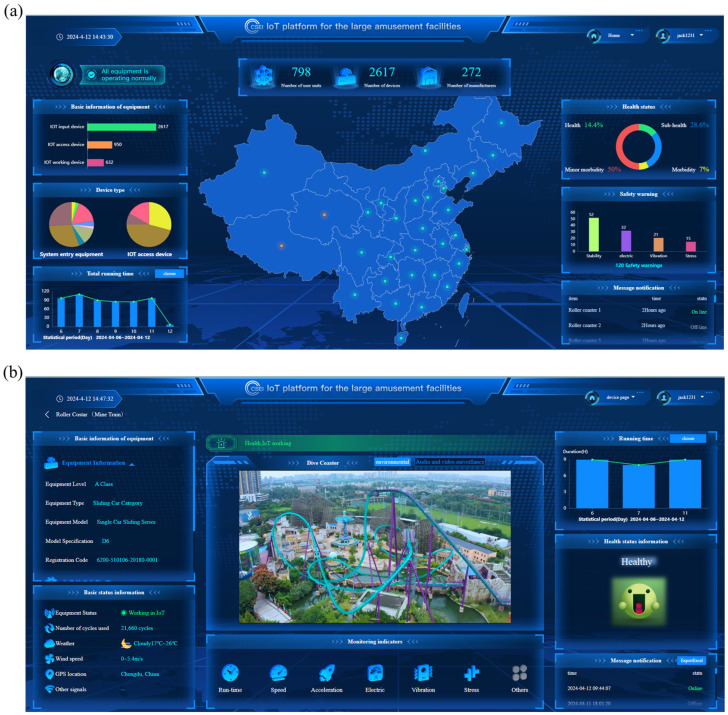
(**a**) The system homepage and (**b**) the equipment monitoring interface within the IoT platform.

**Table 1 sensors-24-04433-t001:** Detailed technical parameters for audio and video surveillance.

Specification	Description
Sensor type	1/2.7″ Progressive Scan CMOS
Maximum resolution	1920 × 1080@25 fps
Supplementary light	White Light/Infrared Dual Supplementary Light
Local storage	256 GB MicroSD Card
Protection rating	IP67
Network interface	RJ45 10 M/100 M Adaptive Ethernet Port
Power supply	DC 12 V/1.08 A

**Table 2 sensors-24-04433-t002:** Detailed technical parameters for photoelectric sensor.

Specification	Description
Response time	600 μs
Output form	NPN
Detection distance	6.5 m
Power supply voltage	10–30 V DC
Detection method	Reflection Plate
Operating temperature	−20 °C~+70 °C

**Table 3 sensors-24-04433-t003:** Detailed technical parameters for acceleration monitoring system.

Specification	Description
Chip model	BMI160
Measurement range	−16~+16 g
Measurement accuracy	0.061 mg/LSB
Sampling frequency	1~200 Hz
Operating temperature	−20 °C~+60 °C
Operating system	Android 8.0
Local storage	10 GB
Communication mode	2.4 G/5 G
Protection level	IPX6
Host size	104 × 68 × 43 mm
Power supply	10,000 mAh detachable lithium-ion battery

**Table 4 sensors-24-04433-t004:** Detailed technical parameters for speed monitoring system.

Specification	Description
Operating frequency	24 GHz
Transmission power	6 dBm
Sampling frequency	20 ms
Speed measurement range	1~220 km/h
Speed measurement accuracy	1 km/h
Beam horizontal angle of view	−24~24 deg
Beam elevation angle of view	−9~9 deg
Operating voltage	AC 90~260 V
Operating temperature	−40~+80 °C
Communication mode	4 G network

**Table 5 sensors-24-04433-t005:** Detailed technical parameters for vibration monitoring system.

Specification	Description
Power supply voltage	DC 10–30 V
Protection class	IP67
Frequency range	10–1600 Hz
Operating temperature	−40~+150 °C
Speed measurement range	0–50 mm/s
Displacement measurement range	0–5000 μm
Acceleration measurement range	±16 g (g = 9.8 m/s^2^)
Acceleration measurement accuracy	<1% (@160 Hz, 10 m/s^2^)
Acceleration display resolution (m/s^2^)	0.1
Communication mode	4G network

**Table 6 sensors-24-04433-t006:** Detailed technical parameters for intelligent circuit breaker.

Specification	Description
Input voltage	AC 220/400 V
Input current	5–600 A
Frequency	45–65 Hz
Operating temperature	−20~+60 °C
Accuracy	0.2%
Power consumption	<1 VA
Power supply mode	Dry Contact
Dimensions	72 × 89 × 59 mm
Communication interface standard	RS485
Energy pulse	Optocoupler Energy Pulse: 80 ms ± 20 ms

**Table 7 sensors-24-04433-t007:** Detailed technical parameters for stress monitoring system.

Specification	Description
Communication protocol	BeeNet
Transmission distance	300 m
Measurement range	±25,000 με
Maximum sampling frequency	1 KSPS
Accuracy class	0.2
A/D precision	24 bits
Local storage	1G
Channel	8
Resolution	±0.8 με
Radio frequency (RF)	2.4G DSSS
Host size	110 × 110 × 30 mm
Power supply	Lithium-ion battery

## Data Availability

Data are contained within the article.

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
