# Peer review of "Development and Application of IoT Monitoring Systems for Typical Large Amusement Facilities"

_sensors, 2024, doi:10.3390/s24144433_

Round 1

Reviewer 1 Report

Comments and Suggestions for Authors

The authors have done some commendable work regarding IoT in roller coaster in an amusement park. The comments to increase the quality of the paper are as follows: 

1. The authors claim in the study to facilitate real-time condition surveillance and fault warnings for roller coasters and improvement in operational safety. However, i didnot found any indices in-terms of quantitated data for the support of their claim. 

2. The novel aspects of the study must be illustrated properly in bulleted form and authors must mention their contribution to the body of knowledge. 

3. There must be a flow chart to illustrated the methodology in proper way. 

4. The authors mentioned Chinese standard GB-8408 (2018), However are heir any other international standards that are compatible? The representative durations (Δt=0.05s, 0.1s, and 0.2s) have considered. What if Δt=0.075s, 0.15s or 0.3s (as thresholds) and sampling interval is a bit less than 0.01s. What will be the impact? Author must mention that in a proper way. 

5. The authors mentioned that they use wireless communication module with IEEE 238 802.15.4 standard-based wireless protocol for data transmission. What is the acceptable latency limit or other concerned impacts. Have you considered them. 

6. Does the authors consider any contingency in speed monitoring system and mounted radar? 

7. The roller coaster  safety factor (n) of 3.5 or above is deemed satisfactory when determining the ratio between the material's ultimate strength and the stress levels recorded at these points. Which material is used and does it change with different metal alloys, please specify that with logical reason? 

8. The research serves more as a data gathering framework and the applications are barely mentioned. Kindly mention them properly as future works. 

9. Where is the reliability improvement which is one of the main objectives of IoT based systems? 

10. On the whole the paper can-be improved in the section of literature review with addition of new relevant references from 2023-2024. Then the authors must mention that how their proposed work is better than the reported literature. 

Author Response

Reviewer #1:

The authors have done some commendable work regarding IoT in roller coaster in an amusement park. The comments to increase the quality of the paper are as follows:

  1. The authors claim in the study to facilitate real-time condition surveillance and fault warnings for roller coasters and improvement in operational safety. However, I didnot found any indices in-terms of quantitated data for the support of their claim.

Our replies:

We thank the reviewer’s constructive comment. We have made corresponding adjustments to the expressions in this paper. This article focuses on the research and development of sensor systems and IoT systems tailored to the characteristics of large-scale amusement facilities, aiming to acquire typical data to provide a data foundation for subsequent fault diagnosis and prediction. However, diagnosis and prediction are not the main content of this article. Therefore, the title is revised to " Development and Application of IoT Monitoring System for Typical Large Amusement Facilities." Currently, the fault warning method adopted in this paper is to issue a fault warning when the monitoring signal deviates from the normal range or does not meet the Chinese standard GB 8408. For example, in the acceleration monitoring described in Section 3.3, we stipulate that the duration of the combined acceleration value should not exceed the three envelope lines in Fig. 5c (Δt = 0.05s, 0.1s, and 0.2s). Apart from utilizing thresholds and relevant standards to assess stable data, a significant technique for fault diagnosis of fluctuating data involves employing artificial intelligence algorithms to detect potential anomalies. For instance, in the case of vibration signals, when the equipment operates normally, the vibration data exhibits fluctuations, rendering it difficult to determine abnormalities through the setting of fixed thresholds.

Our revisions:

In the Abstract, we have made the following adjustments:
This study aims to develop customized operational status monitoring sensors and an IoT platform for large roller coasters, encompassing the design and fabrication of sensors and IoT platforms, data acquisition, and processing. The ultimate objective is to enable timely warnings when monitoring signals deviate from normal ranges or violate relevant standards, thereby facilitating the prompt identification of potential safety hazards and equipment faults.

In the Introduction, we have made the following adjustments:

Therefore, our proposed sensor network and IoT platform facilitate the early detection of abnormal signals in roller coaster operations that deviate from normal ranges or violate national or regional standards. This enables the prompt implementation of maintenance and repair measures to ensure the safety of roller coaster operations.

In the conclusion, we added:

In the forthcoming work, the assessment of data through thresholds and applicable standards will be complemented by the employment of artificial intelligence algorithms as a pivotal technique for identifying potential anomalies in fluctuating data, thereby enhancing the fault diagnosis process.

The title is revised to " Development and Application of IoT Monitoring System for Typical Large Amusement Facilities" in this paper.

  1. The novel aspects of the study must be illustrated properly in bulleted form and authors must mention their contribution to the body of knowledge.

Our replies:

We thank the reviewer’s constructive comment. In the introduction, we have added the novel aspects of the study in bulleted form.

Our revisions:

In Introduction, we added:

The innovations of this paper are as follows:

  • Identification of the required signal types for monitoring the operational status of roller coasters, based on a comprehensive analysis of numerous past failure cases.
  • Development of a specialized monitoring sensor system tailored to the unique struc-tural and operational characteristics of roller coasters.
  • Design of an informative IoT platform for large-scale amusement facilities, featuring the visualization of monitoring data to enhance operational safety and efficiency.

  1. There must be a flow chart to illustrated the methodology in proper way. 

Our replies:

We thank the reviewer’s constructive comment. We have added a flow chart to illustrated the methodology, as shown in Figure 2a.

Our revisions:

Please see a flow chart in Figure 2a in the revised manuscript.

  1. The authors mentioned Chinese standard GB-8408 (2018), However are heir any other international standards that are compatible? The representative durations (Δt=0.05s, 0.1s, and 0.2s) have considered. What if Δt=0.075s, 0.15s or 0.3s (as thresholds) and sampling interval is a bit less than 0.01s. What will be the impact? Author must mention that in a proper way.

Our replies:

We thank the reviewer’s constructive comment. Regarding the consideration of passenger acceleration tolerance, GB-8408 (2018) is compatible with ASTM F2291-2019, EN13814-2019, and ISO/TS 17929-2014. For instance, the representative durations of the allowable combined acceleration values (Δt = 0.05s, 0.1s, and 0.2s) in Figure 5(c) are consistent with EN13814-2019. The Δt values are derived from experimental data, and other Δt values are not specified in the standard. To determine whether the acceleration of an amusement ride meets the standard requirements during operation, it is necessary to identify the duration of different accelerations based on the acceleration history. Due to the large number of sampling points, a simplified acceleration discrimination method is proposed here: first, the acceleration is sampled, with the sampling interval generally defined as Δt ≤ 0.05s (0.01s is selected in this paper), and the state value points (az, ay) of acceleration at different moments during operation are marked in Figure 5(c). The line connecting adjacent points represents the change in acceleration state over Δt. From Figure 5c, the criteria for combined acceleration are as follows:

If (n1+1) Δt > 0.2, the acceleration does not meet the standard requirements.

If (n 2+1) Δt > 0.1, the acceleration does not meet the standard requirements.

If (n 3+1) Δt > 0.05, the acceleration does not meet the standard requirements.

Otherwise, the acceleration meets the standard requirements. Therefore, when the sampling interval Δt is less than 0.01s, more points exceeding the limits are allowed, as long as the above criteria are met.

Our revisions:

Please refer to the corresponding modifications in Section 3.3.

  1. The authors mentioned that they use wireless communication module with IEEE 238 802.15.4 standard-based wireless protocol for data transmission. What is the acceptable latency limit or other concerned impacts. Have you considered them. 

Our replies:

We thank the reviewer’s constructive comment. When the acceleration module transmits data using the wireless protocol based on the IEEE 802.15.4 standard, the acceptable latency limit depends on the specific application scenario. For example, in real-time responsive applications such as acceleration detection for autonomous vehicles, the latency must be extremely low. However, for the roller coaster acceleration monitoring discussed in this paper, the latency tolerance is higher. This is because the acceleration parameters primarily evaluate track abnormalities such as excessive wear and deformation, as well as anomalies in the wheel structure. Typically, data from the previous round is analyzed after the vehicle arrives at the station to determine whether to continue operation. Since the emergency braking position for roller coasters is usually located at the return platform, there is no need for us to immediately respond to acceleration abnormalities with braking. Currently, the latency of data transmission in this paper has no impact.

Our revisions:

Please refer to the corresponding modifications in Section 3.3.

  1. Does the authors consider any contingency in speed monitoring system and mounted radar?

Our replies:

We thank the reviewer’s constructive comment. Firstly, the radar installation position maintains a safe distance from the vehicle operation, ensuring that the detachment of the speed monitoring system poses no risk to the equipment. Furthermore, the speed sensor is used to monitor the vehicle's speed at specific locations. If an abnormal speed is detected, the next round of departures can be halted after the vehicle enters the station, allowing for inspection of the vehicle and track to prevent any potential accidents.

  1. The roller coaster safety factor (n) of 3.5 or above is deemed satisfactory when determining the ratio between the material's ultimate strength and the stress levels recorded at these points. Which material is used and does it change with different metal alloys, please specify that with logical reason?

Our replies:

We thank the reviewer’s constructive comment. For stress monitoring, the main load-bearing components of the car body, track, and column structure are typically the focus of attention. These structures are generally made of carbon steel, such as Q355 steel and 20# steel. Since there is little difference in mechanical properties between these steel grades, the same safety factor of 3.5 is selected according to the requirements of the Chinese standard GB8408-2018.

Our revisions:

Please refer to the corresponding modifications in Section 3.7.

  1. The research serves more as a data gathering framework and the applications are barely mentioned. Kindly mention them properly as future works. 

Our replies:

We thank the reviewer’s constructive comment. In terms of application, after the collected data is fed back to the IoT platform, it can be used to monitor the operational status information of all IoT devices deployed in the amusement park online, as shown in Fig. 10. This study selected Happy Valley in Chengdu, China, as a pilot site and conducted online monitoring applications on four large roller coasters in the park: Dive coaster (Bolliger & Mabillard Consulting Engineers Inc.), Wooden coaster (Great Coasters International, Inc.), MEGA coaster (Intamin Transportation Ltd.), and Mine Train (Vekoma Rides Manufacturing Bv). When the sensor monitoring data experiences abnormal fluctuations or fails to meet relevant standards, the equipment health status in Fig. 10(b) will change and push corresponding fault information.

Our revisions:

Please refer to the corresponding modifications in Section 3.8.

  1. Where is the reliability improvement which is one of the main objectives of IoT based systems?

Our replies:

We thank the reviewer’s constructive comment. In the Sections 3.1 to 3.7 of this paper, a total of seven types of sensors were employed to monitor the reliability of equipment operation in real-time based on seven sets of data. When the monitoring signals exceed the normal fluctuation range or fail to meet the standard requirements, the IoT platform will push information and alerts to users based on the time and location of the abnormal data. For instance, when the stress and acceleration do not comply with the regulations specified in GB-8408, or there are abnormal increases and decreases in vibration signals and electrical signals, the IoT platform will issue reminders for abnormal data, thereby assisting personnel in inspecting the equipment to enhance its reliability.

Our revisions:

Please refer to the corresponding modifications in Section 3.8.

  1. On the whole the paper can-be improved in the section of literature review with addition of new relevant references from 2023-2024. Then the authors must mention that how their proposed work is better than the reported literature.

Our replies:

We thank the reviewer’s constructive comment. We have improved the literature review with revisions highlighted in red in the Introduction. Three references from 2023-2024 in the revised manuscript have been added. Additionally, we have highlighted the advantages of this study compared to previous research.

Our revisions:

In Introduction, we added:

For example, Singh [26] has introduced an intelligent water quality monitoring framework that incorporates AI sensors to predict indicators without the need for electronic sensors, especially employing machine learning algorithms to forecast the concentration of E. coli in water. In addition, a novel machine learning-based IoT system has been developed to track an individual's stress level and movement velocity during physical activities by examining factors such as body temperature and sweat [27]. Capitalizing on wireless sensor networks, machine learning techniques, and IoT methodologies, the detection and classification of pollutants in aquatic environments have been effectively executed [28].

Currently, due to the unique complex structure and diverse operational characteristics of amusement facilities, a dedicated IoT monitoring system for them has not yet been developed.

[26] Singh, Y.; Walingo, T. Smart Water Quality Monitoring with IoT Wireless Sensor Networks. Sensors, 2024, 24(9), 2871.

[27] Al-Atawi, A.A.; Alyahyan, S.; Alatawi, M.N.; Sadad, T.; Manzoor, T.; Farooq-i-Azam, M.; Khan, Z.H. Stress Monitoring Using Machine Learning, IoT and Wearable Sensors. Sensors, 2023, 23(21), 8875.

[28] Da Silva, Y.F.; Freire, R.C.S.; Neto, J.V.D. Conception and Design of WSN Sensor Nodes Based on Machine Learning, Embedded Systems and IoT Approaches for Pollutant Detection in Aquatic Environments. IEEE Access, 2023, 11, 117040-117052.

Reviewer 2 Report

Comments and Suggestions for Authors

This paper analyses large roller coasters, encompassing the examination of equipment monitoring metrics, data acquisition, and processing, with the ultimate objective of fashioning a condition monitoring framework for these rides. This study is to facilitate real-time condition surveillance and fault warnings for substantial roller coasters, thereby substantially bolstering their operational safety.

This work is intresting, and I have the following concerns.

1, How are the employed sensors connected to the central platform? wired or wireless? and why?

2, How do the IoT platform use the sensing data to make decisions? only according to each sensor, or by combining multiple sensors data to detect one problem?

3, How about the related works?

Comments on the Quality of English Language

readable

Author Response

Reviewer #2:

This paper analyses large roller coasters, encompassing the examination of equipment monitoring metrics, data acquisition, and processing, with the ultimate objective of fashioning a condition monitoring framework for these rides. This study is to facilitate real-time condition surveillance and fault warnings for substantial roller coasters, thereby substantially bolstering their operational safety.

This work is interesting, and I have the following concerns.

  1. How are the employed sensors connected to the central platform? wired or wireless? and why?

Our replies:

We thank the reviewer’s constructive comment. In this paper, all sensor connections to the IoT platform are wireless, encompassing both WiFi and 4G modalities. This choice is attributed to the intricate and rapidly changing dynamics of roller coaster movements, wherein wired transmission could potentially interfere with the coaster's operation, thereby compromising passenger and equipment safety.

Our revisions:

In the Introduction, we added:
Due to the complex and rapid changes in roller coaster movements, all sensor connections to the IoT platform in this study are wireless to ensure passenger and equipment safety.

  1. How do the IoT platform use the sensing data to make decisions? only according to each sensor, or by combining multiple sensors data to detect one problem?

Our replies:

We thank the reviewer’s constructive comment. When the monitoring signals exceed the normal fluctuation range or fail to meet the standard requirements (Chinese standard GB8408-2018), the IoT platform will push information and alerts to users based on the time and location of the abnormal data. For instance, when the stress and acceleration do not comply with the regulations specified in GB-8408, or there are abnormal increases and decreases in vibration signals and electrical signals, the IoT platform will issue reminders for abnormal data, thereby assisting personnel in inspecting the equipment to enhance its reliability.

Additionally, we have added a flow chart in Figure 2a. Damage, deformation, rust and wear can be detected by multiple sensors data, including speed and acceleration. Regarding other problems, they are individually reflected by the corresponding single sensor data.

Our revisions:

Please refer to the corresponding modifications in Section 3.8 and the newly added Figure 2a.

  1. How about the related works? 

Our replies:

We thank the reviewer’s constructive comment. We have improved the literature review with revisions highlighted in red in the Introduction. Three references in the revised manuscript have been added. Additionally, we have highlighted the advantages of this study compared to previous research.

Our revisions:

In Introduction, we added:

For example, Singh [26] has introduced an intelligent water quality monitoring framework that incorporates AI sensors to predict indicators without the need for electronic sensors, especially employing machine learning algorithms to forecast the concentration of E. coli in water. In addition, a novel machine learning-based IoT system has been developed to track an individual's stress level and movement velocity during physical activities by examining factors such as body temperature and sweat [27]. Capitalizing on wireless sensor networks, machine learning techniques, and IoT methodologies, the detection and classification of pollutants in aquatic environments have been effectively executed [28].

Currently, due to the unique complex structure and diverse operational characteristics of amusement facilities, a dedicated IoT monitoring system for them has not yet been developed.

[26] Singh, Y.; Walingo, T. Smart Water Quality Monitoring with IoT Wireless Sensor Networks. Sensors, 2024, 24(9), 2871.

[27] Al-Atawi, A.A.; Alyahyan, S.; Alatawi, M.N.; Sadad, T.; Manzoor, T.; Farooq-i-Azam, M.; Khan, Z.H. Stress Monitoring Using Machine Learning, IoT and Wearable Sensors. Sensors, 2023, 23(21), 8875.

[28] Da Silva, Y.F.; Freire, R.C.S.; Neto, J.V.D. Conception and Design of WSN Sensor Nodes Based on Machine Learning, Embedded Systems and IoT Approaches for Pollutant Detection in Aquatic Environments. IEEE Access, 2023, 11, 117040-117052.

Round 2

Reviewer 2 Report

Comments and Suggestions for Authors

In the authors' response, the authors reply that all sensor connections to the IoT platform are wireless, encompassing both Wi-Fi and 4G modalities. On this point, I still have the following concerns.

1, How the wireless sensor networks are established? For example, how about the network architecture, the network topologies? How many Wi-Fi APs are employed? Why are some sensors connected with Wi-Fi, while the others using 4G? Based on what kind of metric to make such decision? Throughput, delay or age of information (AoI)? Or, the types of sensors, the importance of the sensor status updates? I believe your considerations and insights from the real application can help to improve the networking technologies. Thus, more details are suggested to provide.

2, When there are many people in the rush hours, can the employed wireless networking technologies bear the sensing data? Since Wi-Fi uses unlicensed ISM frequency, when the users are dense its performance may degrade sharply. Even for the 4G, its performance may not be good enough in rush hours. However, in the rush hours, the importance of the monitoring system should become more.

3, How often does each sensor report its status update? Based on what kind of metric? For example, the importance of the sensors, or the possible resulted failures? Why? Can the employed networking technologies, Wi-Fi and 4G, meet these requirements?

4, How about the cost of this system? For example, the energy, the money.

Comments on the Quality of English Language

It is readable.

Author Response

We would like to express our sincere appreciation to the reviewer. The reviewer’s comments are all valuable and very helpful for revising and improving our paper.  All revisions are marked in red in the main text. The main corrections and responses to the reviewer’s comments are listed as follows.

Reviewer #2:

In the authors' response, the authors reply that all sensor connections to the IoT platform are wireless, encompassing both Wi-Fi and 4G modalities. On this point, I still have the following concerns.

  1. How the wireless sensor networks are established? For example, how about the network architecture, the network topologies? How many Wi-Fi APs are employed? Why are some sensors connected with Wi-Fi, while the others using 4G? Based on what kind of metric to make such decision? Throughput, delay or age of information (AoI)? Or, the types of sensors, the importance of the sensor status updates? I believe your considerations and insights from the real application can help to improve the networking technologies. Thus, more details are suggested to provide.

Our replies:

We thank the reviewer’s constructive comment. For wireless sensors using Wi-Fi networking, local communication is conducted via TCP/IP. Depending on the location of the sensors, Wi-Fi access points are configured nearby to ensure that all wireless sensors are within the same local area network (LAN). Since the sensors need to be connected to a data acquisition terminal, which has a limited WIFI signal range and requires nearby network access, we need to deploy a Wi-Fi access point within no more than 30 meters. The sensor signals within the LAN are collected through Data Acquisition and then uniformly sent to the cloud server on the internet via the Processor Terminal. For sensors with 4G communication capability, they will directly communicate with the cloud server deployed on the Internet.

The choice between 4G and Wi-Fi depends on the amount of data collected by the sensors. For photoelectric sensors, which collect switching signals, 4G can be directly used to upload the data to the cloud. However, for vibration acceleration monitoring data, which requires local data processing terminals for analysis and processing, the data volume is larger. Using 4G transmission would result in higher costs, therefore, Wi-Fi is selected for data packaging and uploading.

Our revisions:

In Section 3, we added:
Wi-Fi-enabled wireless sensors communicate locally via TCP/IP, forming a unified LAN with nearby access points. The sensor signals within the LAN are collected through Data Acquisition and then uniformly sent to the cloud server on the internet via the Processing Terminal. 4G-capable sensors communicate directly with the cloud. Please refer to the corresponding modifications on page 4.

  1. When there are many people in the rush hours, can the employed wireless networking technologies bear the sensing data? Since Wi-Fi uses unlicensed ISM frequency, when the users are dense its performance may degrade sharply. Even for the 4G, its performance may not be good enough in rush hours. However, in the rush hours, the importance of the monitoring system should become more.

Our replies:

We thank the reviewer’s constructive comment. We utilize the 5GHz frequency band for Wi-Fi wireless networks. The 5GHz frequency band boasts more channels and wider bandwidth, thus enhancing the network's carrying capacity. The data undergoes preprocessing at the Data Acquisition stage, including compression and packaging. On the other hand, considering the design from the user's perspective, the data is presented based on the operational rounds of the amusement equipment, meaning that the various data for the current round are displayed only after the completion of each round. Therefore, the system has a relatively high tolerance for latency. Even if there may be delays in the network during peak hours, it can still ensure that the data is displayed on the terminal after the completion of the current round.

Our revisions:

In Section 3.8, we added:

Utilizing the 5GHz Wi-Fi band, this study enhances network capacity during peak hours through its increased channels and bandwidth. The monitoring data is presented based on the operational rounds of the roller coaster, ensuring data display for each round only after completion. This approach grants the system a relatively high tolerance for latency, enabling data to be showcased on terminals post-round completion despite potential delays during peak periods. Please refer to the corresponding modifications on page 17.

  1. How often does each sensor report its status update? Based on what kind of metric? For example, the importance of the sensors, or the possible resulted failures? Why? Can the employed networking technologies, Wi-Fi and 4G, meet these requirements?

Our replies:

We thank the reviewer’s constructive comment. Based on the different test data, sensors have varying upload frequencies. For physical parameters that may lead to short-term failures in roller coasters, the frequency of data upload by sensors will be relatively high. During the operation of the equipment, sensors such as electrical signals, vibration, and acceleration upload data once per second. For instance, sensors monitoring operating speed will upload data when a detection event occurs. The system employs IoT communication technology based on the MQTT protocol, ensuring data upload under weak network conditions. Currently, the Wi-Fi and 4G network communication used can meet the requirements for sensor data transmission in roller coaster operation monitoring.

Our revisions:

Please refer to the corresponding modifications on page 17.

  1. How about the cost of this system? For example, the energy, the money.

Our replies:

We thank the reviewer’s constructive comment. In terms of energy costs, the total operating power of all sensors, processing terminals, and network equipment discussed in this paper does not exceed 200W.

To effectively reduce hardware costs and integrate the characteristics of amusement facilities, we have developed a dedicated sensor system encompassing speed monitoring, acceleration monitoring, electrical signal monitoring, and more. The specific hardware cost needs to be comprehensively considered based on the type, quantity, and location of the sensors deployed in different amusement facilities. Regarding software costs, the system is deployed on a cloud server, which has significant advantages in terms of initial investment, routine maintenance, elastic scalability, and indirect costs. This helps amusement parks reduce overall costs, and the more roller coasters that are connected to IoT, the lower the average cost becomes.